# Stylus:
# Automatic Adapter Selection for Diffusion Models

**Michael Luo**[1]    **Justin Wong**[1]    **Brandon Trabucco**[2]    **Yanping Huang**[3]
**Joseph E. Gonzalez**[1]    **Zhifeng Chen**[3]    **Ruslan Salakhutdinov**[2]    **Ion Stoica**[1]
[1]UC Berkeley    [2]CMU MLD    [3]Google Deepmind
{michael.luo,wong.justin,jegonzal,istoica}@berkeley.edu
{btrabucc,rsalakhu}@cs.cmu.edu
{huangyp,zhifengc}@google.com

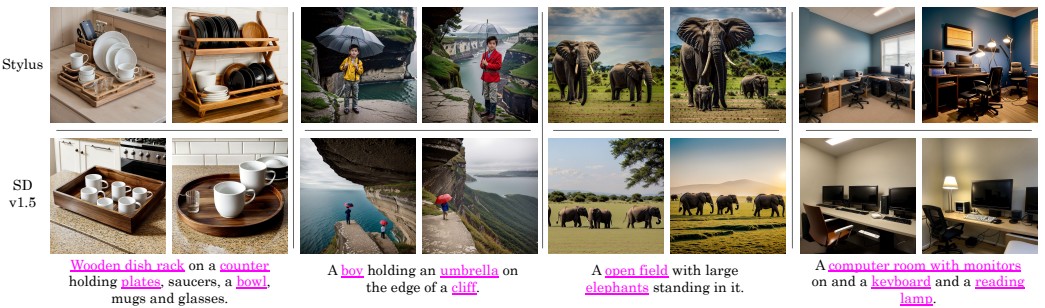

Figure 1: **Adapter Selection.** Given a user-provided prompt, our method identifies highly relevant adapters (e.g. Low-Rank Adaptation, LoRA) that are closely aligned with the prompt's context and at least one of the prompt's keywords. Composing relevant adapters into Stable Diffusion improves visual fidelity, image diversity, and textual alignment. Note that these prompts are sampled from MS-COCO [22].

## Abstract

Beyond scaling base models with more data or parameters, fine-tuned adapters provide an alternative way to generate high fidelity, custom images at reduced costs. As such, adapters have been widely adopted by open-source communities, accumulating a database of over 100K adapters—most of which are highly customized with insufficient descriptions. To generate high quality images, This paper explores the problem of matching the prompt to a *set* of relevant adapters, built on recent work that highlight the performance gains of composing adapters. We introduce Stylus, which efficiently selects and automatically composes task-specific adapters based on a prompt's keywords. Stylus outlines a three-stage approach that first summarizes adapters with improved descriptions and embeddings, retrieves relevant adapters, and then further assembles adapters based on prompts' keywords by checking how well they fit the prompt. To evaluate Stylus, we developed `StylusDocs`, a curated dataset featuring 75K adapters with pre-computed adapter embeddings. In our evaluation on popular Stable Diffusion checkpoints, Stylus achieves greater CLIP/FID Pareto efficiency and is twice as preferred, with humans and multimodal models as evaluators, over the base model. See stylus-diffusion.github.io for more.

## 1 Introduction

In the evolving field of generative image models, finetuned adapters [7, 11] have become the standard, enabling custom image creation with reduced storage requirements. This shift has spurred the growth of extensive open-source platforms that encourage communities to develop and share different

38th Conference on Neural Information Processing Systems (NeurIPS 2024).

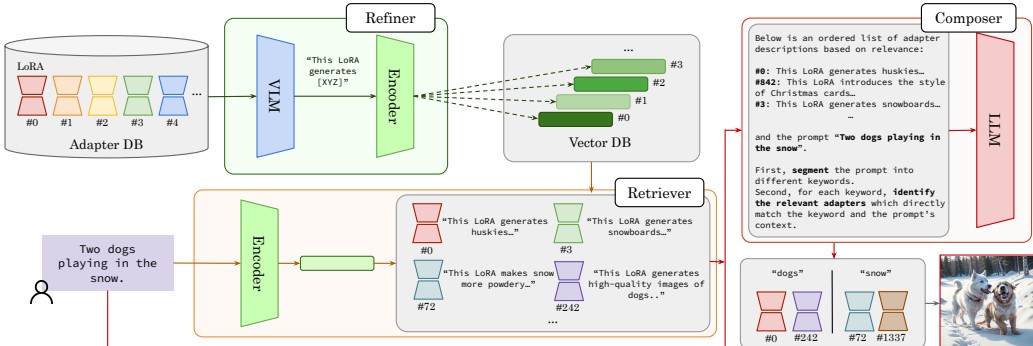

Figure 2: **Stylus algorithm.** Stylus consists of three stages. The *refiner* plugs an adapter's model card through a VLM to generate textual descriptions of an adapter's task and then through an encoder to produce the corresponding text embedding. The *retriever* fetches candidate adapters that are relevant to the entire user prompt. Finally, the *composer* prunes and jointly categorizes the remaining adapters based on the prompt's tasks, which correspond to a set of keywords.

adapters and model checkpoints, fueling the proliferation of creative AI art [28, 51]. As the ecosystem expands, the number of adapters has grown to over 100K, with Low-Rank Adaptation (LoRA) [14] emerging as the dominant finetuning approach (see Fig. 3). A new paradigm has emerged where users manually select and creatively compose multiple adapters, on top of existing checkpoints, to generate high-fidelity images, moving beyond the standard approach of improving model class or scale.

In light of performance gains, our paper explores the automatic selection of adapters based on user-provided prompts (see Fig. 1). However, selecting relevant adapters presents unique challenges compared to existing retrieval-based systems, which rank relevant texts via lookup embeddings [18]. Specifically, efficiently retrieving adapters requires converting adapters into lookup embeddings, a step made difficult with low-quality documentation or no direct access to training data—a common issue on open-source platforms. Furthermore, in the context of image generation, user prompts often imply multiple highly-specific tasks. For instance, the prompt "two dogs playing the snow" suggests that there are two tasks: generating images of "dogs" and "snow". This necessitates segmenting the prompt into various tasks (i.e. keywords) and selecting relevant adapters for each task, a requirement beyond the scope of existing retrieval-based systems [9]. Finally, composing multiple adapters can degrade image quality, override existing concepts, and introduce unwanted biases into the model (see App. A.4).

We propose Stylus, a system that efficiently assesses user prompts to retrieve and compose sets of highly-relevant adapters, automatically augmenting generative models to produce diverse sets of high quality images. Stylus employs a three-stage framework to address the above challenges. As shown in Fig. 2, the *refiner* plugs in an adapter's model card, including generated images and prompts, through a multi-modal vision-language model (VLM) and a text encoder to pre-compute concise adapter descriptions as lookup embeddings. Similar to prior retrieval methods [18], the *retriever* scores the relevance of each embedding against the user's entire prompt to retrieve a set of candidate adapters. Finally, the *composer* segments the prompt into disjoint tasks, further prunes irrelevant candidate adapters, and assigns the remaining adapters to each task. We show that the composer identifies highly-relevant adapters and avoids conceptually-similar adapters that introduce biases detrimental to image generation (§ 4.3). Finally, Stylus applies a binary mask to control the number of adapters per task, ensuring high image diversity by using different adapters for each image and mitigating challenges with composing many adapters.

To evaluate our system, we introduce `StylusDocs`, an adapter dataset consisting of 75K LoRAs[1], that contains pre-computed adapter documentations and embeddings from Stylus's *refiner*. Our results demonstrate that Stylus improves visual fidelity, textual alignment, and image diversity over popular Stable Diffusion (SD 1.5) checkpoints—shifting the CLIP-FID Pareto curve towards greater efficiency and achieving up to 2x higher preference scores with humans and vision-language models (VLMs) as evaluators. As a system, Stylus is practical and does not present large overheads to the batch image generation process. Finally, Stylus can extend to different image-to-image application domains, such as image inpainting and translation.

---

[1]Sourced from `https://civitai.com/` [28].

## 2 Related Works

**Adapters.** Adapters efficiently fine-tune models on specific tasks with minimal parameter changes, reducing computational and storage requirements while maintaining similar performance to full fine-tuning [7, 11, 14].

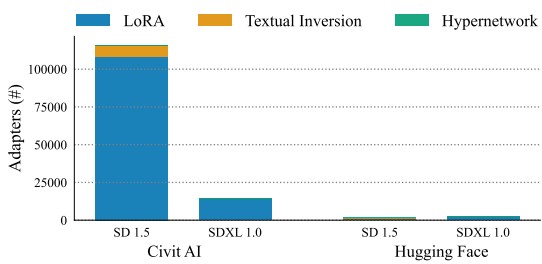

Figure 3: **Number of Adapters.** Civit AI boasts 100K+ adapters for Stable Diffusion, outpacing that of Hugging Face. Low-Rank Adaptation (LoRA) is the dominant approach for finetuning.

Our study focuses on retrieving and merging multiple Low-Rank adapters (LoRA), the popular approach within existing open-source communities [28, 29, 51].

Adapter composition has emerged as a crucial mechanism for enhancing the capabilities of foundational models across various applications [19, 36, 40, 45, 46]. For large language models (LLM), the linear combination of multiple adapters improves in-domain performance and cross-task generalization [3, 15, 16, 48, 49, 55]. In the image domain, merging LoRAs effectively composes different tasks—concepts, characters, poses, actions, and styles—together, yielding images of high fidelity that closely align with user specifications [25, 56]. Adapters also play a key role in synthetic data methods in few-shot computer vision [47]. Our approach advances this further by actively segmenting user prompts into distinct tasks and merging the appropriate adapters for each task.

**Retrieval-based Methods.** Retrieval-based methods, such as retrieval-augmented generation (RAG), significantly improve model responses by adding semantically similar texts from a vast external database [18]. These methods convert text to vector embeddings using text encoders, which are then ranked against a user prompt based on similarity metrics [4, 9, 21, 27, 37, 39]. Similarly, our work draws inspiration from RAG to encode adapters as vector embedings: leveraging visual-language foundational models (VLM) to generate semantic descriptions of adapters, which are then translated into embeddings.

A core limitation to RAG is limited precision, retrieving semi-relevant documents that do not exactly answer the prompt. This leads to a "needle-in-the-haystack" problem, where more relevant documents are buried further down the list [9]. Recent work introduce *reranking* step; this technique uses cross-encoders to assess both the raw user prompt and the ranked set of raw texts individually, thereby discovering texts based on actual relevance [27, 38]. Rerankers have been successfully integrated with various LLM-application frameworks [2, 24, 35].

## 3 Our Method: Stylus

Adapter selection presents three distinct challenges compared to existing methods for retrieving text documents, as outlined in Section 2. First, computing embeddings for adapters is a novel task, made more difficult without access to training datasets. Furthermore, in the context of image generation, user prompts often specify multiple highly fine-grained tasks. This challenge extends beyond retrieving relevant adapters relative to the entire user prompt, but also matching them with specific tasks within the prompt. Finally, composing multiple adapters can degrade image quality and inject foreign biases into the model. Our three-stage framework below—**R**efine, **R**etrieve, and **C**ompose—addresses the above challenges (Fig. 2).

### 3.1 Refiner

The *refiner* is a two-stage pipeline designed to generate textual descriptions of an adapter's task and the corresponding text embeddings for retrieval purposes. This approach is analogous to pre-computed embeddings over an external database of texts in retrieval-based methods [18].

Given an adapter $A_i$, the first stage is a vision-language model (VLM) that takes in the adapter's model card—a set of randomly sampled example images from the model card $\mathcal{I}_i \in \{I_{i1}, I_{i2}, ...\}$, the corresponding prompts $\mathcal{P}_i \in \{p_{i1}, p_{i2}, ...\}$, and an author-provided description,[2] $D_i$—and returns an improved description $D_i^*$. Optionally, the VLM also recommends the weight for LoRA-based adapters,

---

[2]We note that a large set of author descriptions are inaccurate, misleading, or absent. The *refiner* helped correct for human errors by using generated images as the ground truth, significantly improving our system.

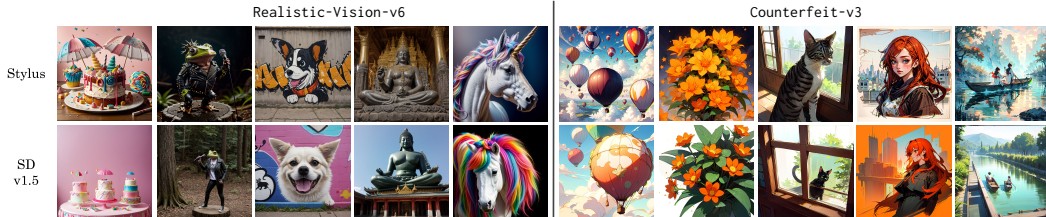

Figure 4: **Qualitative comparison between Stylus over realistic (left) and cartoon (right) style Stable Diffusion checkpoints.** Stylus produces highly detailed images that correctly depicts keywords in the context of the prompt. For the prompt "A graffiti of a corgi on the wall", our method correctly depicts a spray-painted corgi, whereas the checkpoint generates a realistic dog.

as the adapter weight is usually specified either in the author's description $D_i$ or the set of prompts $P_i$, a feature present in popular image generation software [1]. We denote this weight/coefficient as $\alpha_i$. If information cannot be found, the LoRA's weight is set to $\alpha_i = 0.8$. In our experiments, these improved descriptions were generated by Gemini Ultra [43] (see § A.1 for prompt). We chose the Gemini class of models since it has mature safety guardrailing. Specifically, Google's VertexAI API provides stringent safety settings to block explicit content for the input prompt. Safety filters helped us filter out around 30% of original adapters that were tagged as non-explicit by other model repositories.

The second stage uses an embedding model ($\mathcal{E}$) to embed the text description $D_i^*$ for each adapters to yield embeddings, $e_i = \mathcal{E}(D^*)$. In our experiments, we create embeddings from OpenAI's `text-embedding-3-large` model [21, 30]. We store pre-computed embeddings in a vector database, formally notated by the matrix, $V$.

## 3.2 Retriever

The *retriever* fetches the most relevant adapters over the entirety of the user's prompt using cosine similarity. Precisely, the retriever employs the same embedding model ($\mathcal{E}$) to process the user prompt, $q$, generating embedding $e_q = \mathcal{E}(q)$. Using the vector database, we calculate exact cosine similarity scores between the prompt's embedding $e_s$ and the embedding of each adapter in the matrix $V$. The similarity vector, $s_q = \frac{e_q^T V}{|e_q||V|}$, scores the adapter descriptions by similarity. The retriever simply returns indices of the top-k adapters $\mathcal{A}_k = \text{top-k}(s_q)$. In our experiments, we find $k = 150$ is effective for StylusDocs. We denote the set of $k$ descriptions of the adapters, $\mathcal{A}_k$ as $D_k^*$.

## 3.3 Composer

The *composer* serves a dual purpose: segmenting the prompt into tasks from a prompt's keywords and assigning retrieved adapters to tasks. This implicitly filters out adapters that are not semantically aligned with the prompt and detects those likely to introduce foreign bias to the prompt through keyword grounding. For example, if the prompt is "pandas eating bamboo", the composer may discard an irrelevant "grizzly bears" adapter and a biased "panda mascots" adapter.

The composer ($\mathcal{C}$) is a function of the prompt ($q$), the top $K$ adapters ($\mathcal{A}_K$) from the retriever. Formally, denote the tasks identified by the composer as $\mathcal{T}(q) = \{t_1, t_2, \ldots, t_n\}$. The composer produces a mapping from task to adapters:

$$\mathcal{C}(s, \mathcal{A}_K) = \{(t_i, \mathcal{A}_{k_i}) \,|\, t_i \in \mathcal{T}(q), \mathcal{A}_{k_i} \subseteq \mathcal{A}_K, \forall j \in \mathcal{A}_{k_i}, Align(\mathcal{A}_j, t_i)\} \tag{1}$$

where $\mathcal{A}_{k_i}$ is the subset of adapters per task $t_i$, $Align(A_j, t_i)$ is a predicate that holds if the adapter, $A_j$ is aligned with the task, $t_i$.

While the composer can be further improved by fine-tuning with human-labeled data [34], we find that prompting a long-context Large Language Model (LLM) suffices. The LLM accepts the adapter descriptions and the prompt as part of its context and returns a mapping of tasks to a curated set of adapters. In practice, the alignment function is determined in the LLM's chain-of-thought procedure before it outputs the final mapping of adapters to tasks. In our implementation, we choose Gemini 1.5, with a 128K context window, as the composer's LLM (see App. A.3 for the full prompt).

Stylus's composer is similar to *reranking*. Rerankers employ cross encoders ($\mathcal{F}$) that compare the retriever's individual adapter descriptions, generated from the refiner, against the user prompt to determine better similarity scores: $\mathcal{F}(p, D^*)$. This prunes for adapters based on semantic relevance, thereby improving search quality, but not over keyword alignment. Our experimental ablations (§ 4.3) show that our composer outperforms existing rerankers (Cohere, `rerank-english-v2.0`) [38].

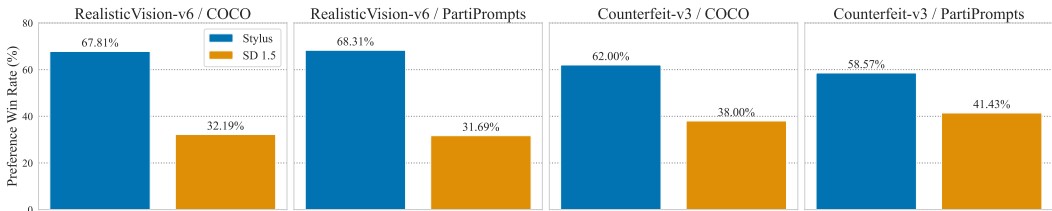

Figure 5: **Human Evaluation.** Stylus achieves a higher preference scores (2:1) over different datasets and Stable Diffusion checkpoints.

## 3.4 Masking

The composer maps tasks to corresponding sets of highly relevant adapters. To further mitigate sensitivity to low-quality adapters, Stylus reduces the number of selected adapters with a straightforward masking scheme. Specifically, for each task, candidate masks are generated, and one is randomly selected to be applied over the set of adapters. Formally, for a given task, $m_i \in \{0,1\}^{|\mathcal{A}_k|}$, is either a one hot encoding, $\vec{1}$, or $\vec{0}$, forming a set of possible masks, $M_i$. Across all tasks, masks are combined by taking the cross-product, $G = M_1 \times M_2 \times ... \times M_n$. The combinatorial sets of masking schemes enable diverse linear combinations of adapters for a single prompt, leading to highly-diverse images (§ 4.2.3). This approach also curtails the number of final adapters merged into the base model, minimizing the risk of composing low-quality adapters that may introduce undesirable effects to the image [56].

## 3.5 Merging

Stylus employs two key insights for effectively merging adapter weights. First, when applied to a single task, large adapter weights can introduce notable visual artifacts, such as over-saturation (Fig. 14a). Second, across multiple tasks, adapters tend to be orthogonal in the weight space, as they are designed to modify distinct, orthogonal concepts [8]. Hence, Stylus computes the final adapter weights by *averaging* weights per task and *summing* weights across tasks. This approach ensures that the adapter weights per task remain appropriately scaled.

We mathematically illustrate our merging scheme below. Recall, the refiner outputs $\alpha_i$, the recommended weight/coefficient, for each adapter. (§ 3.1). As shown in recent work [56], multiple LoRAs can be merged with the base model weights ($W_{base}$). We arrive at our final merged model weights by a summing the adapter weights normalized by task. For a mapping, $\{(t_1, \mathcal{A}_{k_1}), (t_2, \mathcal{A}_{k_2}), \ldots (t_n, \mathcal{A}_{k_n})\}$, and $g = (m_1, m_2, \ldots, m_n) \in G$, the final model weight is:

$$W' = W_{base} + \beta \cdot \sum_{i \le n} \sum_{j \in x_i} \alpha_j \Delta_j / |x_i| \qquad (2)$$

where $x_i = Mask(m_i, \mathcal{A}_i)$ and $\Delta_j$ is the LoRA's weight. We set $\beta = 0.8$ to mitigate image saturation, where assigning high adapter weights to an individual task (or concept) leads to sharp decreases in image quality (see App. A.4). For batch inference, Stylus returns images sorted by CLIP score.

## 4 Results

### 4.1 Experimental Setup

**Adapter Testbed.** Adapter selection requires a large database of adapters to properly evaluate its performance. However, existing methods [15, 55] only evaluate against 50-350 adapters for language-based tasks, which is insufficient for our use case, since image generation relies on highly fine grained tasks that span across many concepts, poses, styles, and characters. To bridge this gap, we introduce `StylusDocs`, a comprehensive dataset that pulls 75K LoRAs from popular model repositories, Civit AI and HuggingFace [28, 51]. This dataset contains precomputed OpenAI embeddings [21] and improved adapter descriptions from Gemini Ultra-Vision [43], the output of Stylus's refiner component (§ 3.1). We further characterize the distribution of adapters in App. A.3.

**Generation Details.** We assess Stylus against Stable-Diffusion-v1.5 [40] as the baseline model. Across experiments, we employ two well-known checkpoints: `Realistic-Vision-v6`, which excels in producing realistic images, and `Counterfeit-v3`, which generates cartoon and anime-style images. Our image generation process integrates directly with Stable-Diffusion WebUI [1] and defaults to 35 denoising steps using the default DPM Solver++ scheduler [26]. To replicate high-quality images from existing users, we enable high-resolution upscaling to generate 1024x1024

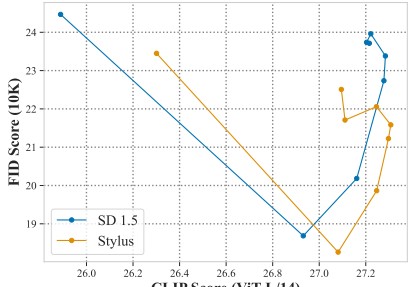

| | CLIP ($\Delta$) | FID ($\Delta$) |
|---|---|---|
| Stylus | **27.25** (+0.03) | **22.05** (-1.91) |
| Reranker | 25.48 (-1.74) | 22.81 (-1.15) |
| Retriever-only | 24.93 (-2.29) | 24.68 (+0.72) |
| Random | 26.34 (-0.88) | 24.39 (+0.43) |
| SD v1.5 | 27.22 | 23.96 |

(a) Clip/FID Pareto Curve for COCO.

(b) CLIP/FID scores and deltas over different retrieval methods (with CFG=6).

Figure 6: **Automatic Evaluation Metrics.** Figure **(a)** plots the CLIP/FID pareto curve. We observe Stylus shifts the curve down (improved visual fidelity, FID) and to the right (improved textual alignment, CLIP score) over a range of guidance values (CFG): [1, 1.5, 2, 3, 4, 6, 9, 12]. Table **(b)** evaluates Stylus against different retrieval methods. Stylus outperforms existing retrieval-based methods, attains the best FID score, and achieves similar CLIP score to Stable Diffusion.

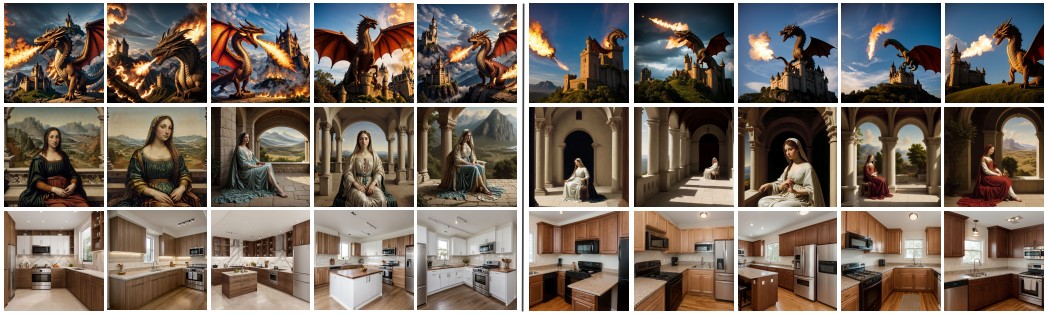

Figure 7: **Image Diversity.** Given the same prompt, our method (left) generates more diverse and comprehensive sets of images than that of existing Stable Diffusion checkpoints (right). Stylus's diversity comes from its masking scheme and the composer LLM's temperature parameter.

from 512x512 images, with the default latent upscaler [17] and denoising strength set to 0.7. For images generated by Stylus, we discovered adapters could shift the image style away from the checkpoint's original style. To counteract this, we introduce a *debias prompt* injected at the end of a user prompt to steer images back to the checkpoint's style[3]. We launched 16 replicas of Stylus and Stable Diffusion on 8 A100-80GB GPUs for 4 weeks to generate images for evaluation.

### 4.2 Main Experiments

#### 4.2.1 Human Evaluation.

To demonstrate our method's general applicability, we evaluate Stylus over a cross product of two datasets, Microsoft COCO [22] and PartiPrompts [53], and two checkpoints, which generate realistic and anime-style images respectively. Examples of images generated in these styles are displayed in Figure 4; Stylus generates highly detailed images that better focus on specific elements in the prompt.

To conduct human evaluation, we enlisted four users to assess 150 images from both Stylus and Stable Diffusion v1.5 for each dataset-checkpoint combination. These raters were asked to indicate their preference for Stylus or Stable-Diffusion-v1.5. In Fig. 5, users generally showed a preference for Stylus over existing model checkpoints. Although preference rates were consistent across datasets, they varied significantly between different checkpoints. Adapters generally improve details to their corresponding tasks (e.g. generate detailed elephants); however, for anime-style checkpoints, detail is less important, lowering preference scores.

#### 4.2.2 Automatic Benchmarks.

We assess Stylus using two automatic benchmarks: CLIP [12], which measures the correlation between a generated images' caption and users' prompts, and FID [13], which evaluates the diversity and aesthetic quality of image sets. We evaluate COCO 2014 validation dataset, with 10K

---

[3]The debias prompts are "realistic, high quality" for `Realistic-Vision-v6` and "anime style, high quality" for `Counterfeit-v3`, respectively.

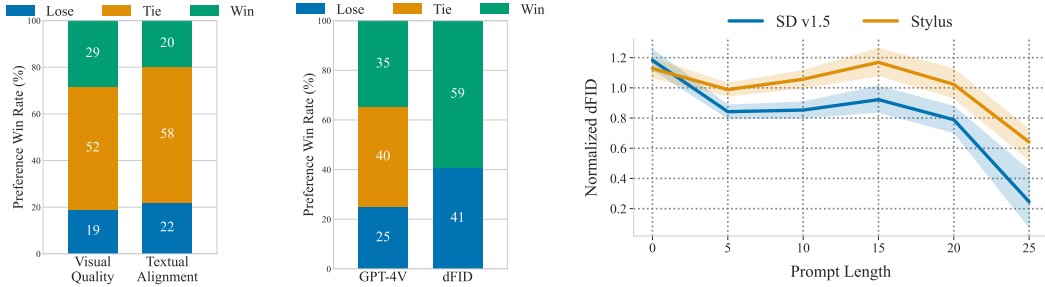

(a) VLM as a Judge with GPT-4V

(b) Image Diversity (GPT-4V, dFID)

(c) Image diversity (*d*FID) versus increasing prompt lengths.

Figure 8: Figure **(a)** and **(b)** evaluate the preference win rate using GPT-4V as a judge. Stylus achieves higher preference scores as judged by GPT-4V for visual quality and image diversity. Figure **(c)** shows that Stylus achieves higher diversity scores than Stable Diffusion whens prompt length increases.

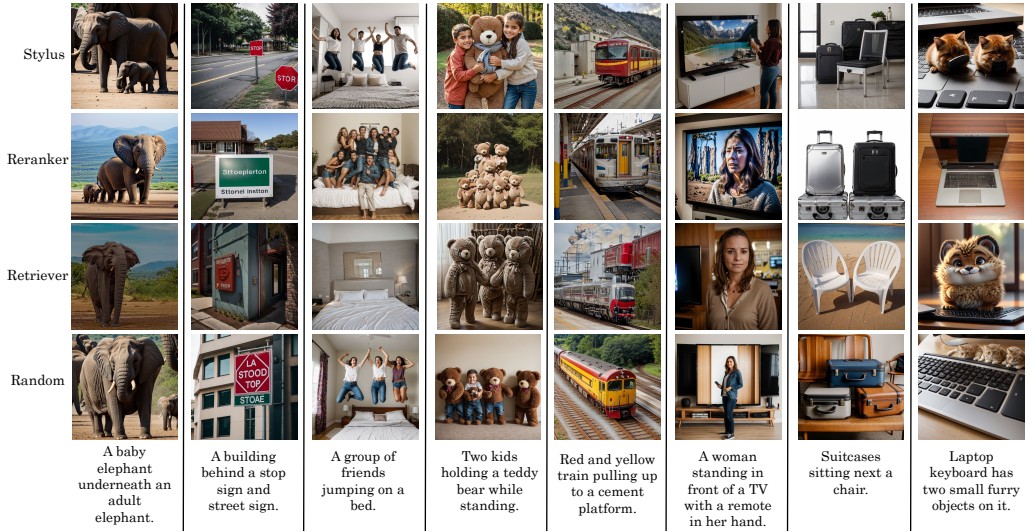

Figure 9: **Different Retrieval Methods.** Stylus outperforms all other retrieval methods, which choose adapters than either introduce foreign concepts to the image or override other concepts in the prompt, reducing textual alignment.

sampled prompts, and the `Realistic-Vision-v6` checkpoint. Fig. 6a shows that Stylus shifts the Pareto curve towards greater efficiency, achieving better visual fidelity and textual alignment. This improvement aligns with our human evaluations, which suggest a correlation between human preferences and the FID scores.

### 4.2.3 VLM as a Judge

We use *VLM as a Judge* to assess two key metrics: textual alignment and visual fidelity, simulating subjective assessments [5]. For visual fidelity, the VLM scores based on disfigured limbs and unrealistic composition of objects. When asked to make subjective judgements, autoregressive models tend to exhibit bias towards the first option presented. To combat this, we evaluate Stylus under both orderings and only consider judgements that are consistent across reorderings; otherwise, we label it a tie. In Fig. 8a, we assess evaluate 100 randomly sampled prompts from the PartiPrompts dataset [53]. Barring ties, we find visual fidelity achieves 60% win rate between Stylus and the Stable Diffusion realistic checkpoint, which is conclusively consistent with the 68% win rate from our human evaluation. For textual alignment, we find negligible differences between Stylus and the Stable Diffusion checkpoint. As most prompts lead to a tie, this indicates Stylus does not introduce additional artifacts. We provide the full prompt in Appendix A.5.

#### 4.2.4 Diversity per Prompt

Given identical prompts, Stylus generates highly diverse images due to different composer outputs and masking schemes. Qualitatively, Fig. 7 shows that Stylus generates dragons, maidens, and kitchens in diverse positions, concepts, and styles. To quantitatively assess this diversity, we use two metrics:

*d*FID: Previous evaluations with FID [13] show that Stylus improves image quality and diversity *across prompts*[4]. We define *d*FID specifically to evaluate diversity per prompt, calculated as the variance of latent embeddings from InceptionV3 [42]. Mathematically, *d*FID involves fitting a Normal distribution $\mathcal{N}(\mu, \Sigma)$ to the latent features of InceptionV3, with the metric given by the trace of the covariance matrix, $d\text{FID} = \text{Tr} \, \Sigma$.

GPT-4V: We use *VLM as a Judge* to assess image diversity between images generated using Stylus and the Stable Diffusion checkpoint over PartiPrompts. Five images are sampled per group, Stylus and SD v1.5, with group positions randomly swapped across runs to avoid GPT-4V's positional bias [56]. Similar to VisDiff, we ask GPT-4V to rate on a scale from 0-2, where 0 indicates no diversity and 2 indicates high diversity [6]. Full prompt and additional details are provided in App A.5.

Fig. 8b displays preference rates and defines a win when Stylus achieves higher *d*FID or receives a higher score from GPT-4V for a given prompt. Across 200 prompts, Stylus prevails in approximately 60% and 58% cases for *d*FID and GPT-4V respectively, excluding ties. Figure 8c compares Stylus with base Stable Diffusion 1.5 across prompt lengths, revealing that Stylus consistently produces more diverse images. Additional results measuring diversity per keyword are presented in Appendix A.6.

### 4.3 Ablations

#### 4.3.1 Impact of Refiner

|  | CLIP ($\triangle$) | FID ($\triangle$) |
|---|---|---|
| No-Refiner | 24.91 (-2.31) | 24.26 (+0.30) |
| Gemini-Ultra Refiner | 27.25 (+0.03) | 22.05 (-1.91) |
| GPT-4o Refiner | 28.04 (+0.82) | 21.96 (-2.00) |
| SD v1.5 | 27.22 | 23.96 |

Figure 10: **Refiner's impact on End2End performance.** Without a refiner, Stylus performs worse than SD v1.5 due to the poor quality of author-provided descriptions. Annotating adapters with GPT-4o significantly improves adapter descriptions and achieves higher CLIP/FID scores than Stylus's default refiner VLM, Gemini-Ultra.

Table 10 evaluates the impact of different refiner pipelines on Stylus's end-to-end performance. Below, we describe each refiner baseline:

**No-Refiner**: Stylus uses baseline adapter descriptions sourced from popular repositories such as Hugging-Face [28, 51]. These descriptions are often low-quality and underspecified. Hence, Stylus chooses the wrong adapters and attains lower CLIP and FID scores relative to SDv1.5.

**Gemini-Ultra Refiner**: This refiner, used throughout all our experiments, employs Gemini-Ultra to auto-generate enhanced adapter descriptions, improving both relevance and specificity. Consequently, Stylus attains better CLIP and FID scores than SDv1.5.

**GPT-4o Refiner**: The GPT-4o refiner, OpenAI's most advanced model, outputs the best adapter descriptions, yielding the highest performance gains across CLIP and FID scores. This baseline demonstrates that Stylus's end-to-end performance is highly dependent on the quality and specificity of adapter descriptions.

#### 4.3.2 Alternative Retrieval-based Methods

We benchmark Stylus's performance relative to different retrieval methods. For all baselines below, we select the top three adapters and merge them into the base model.

**Random**: Adapters are randomly sampled without replacement from StylusDocs.

**Retriever**: The retriever emulates standard RAG pipelines [18, 55], functionally equivalent to Stylus without the composer stage. Top adapters are fetched via cosine similarity over adapter embeddings.

**Reranker**: An alternative to Stylus's composer, the reranker fetches the retriever's adapters and plugs a cross-encoder that outputs the semantic similarity between adapters' descriptions and the prompt. We evaluate with Cohere's reranker endpoint [38].

As shown in Tab. 6b, Stylus achieves the highest CLIP and FID scores, outperforming all other baselines which fall behind the base Stable Diffusion model. First, both the retriever and reranker

---

[4]FID fails to disentangle image fidelity from diversity [32, 41].

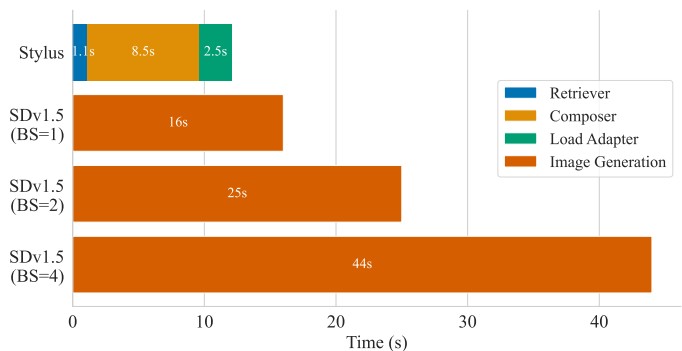

Figure 11: **Comparison of Stylus's inference overheads with Stable Diffusion's inference time by batch size (BS).** At BS=1, Stylus accounts for $75\%$ of the image generation time, primarily due to the composer processing long context prompts from adapter descriptions. However, Stylus's overhead decreases when batch size increases.

significantly underperform compared to Stable Diffusion. Each method selects adapters that are *similar* to the prompt but potentially introduce unrelated biases. In Fig. 9, both methods choose adapters related to elephant movie characters, which biases the concept of elephants and results in depictions of unrealistic elephants. Furthermore, both methods incorrectly assign weights to adapters, causing adapters' tasks to overshadow other tasks within the same prompt. In Fig. 9, both the reranker and retriever generate images solely focused on singular items—beds, chairs, suitcases, or trains—while ignoring other elements specified in the prompt. We provide an analysis of failure modes in A.4.

Conversely, the random policy exhibits performance comparable, but slightly worse, to Stable Diffusion. The random baseline chooses adapters that are orthogonal to the user prompt. Thus, these adapters alter unrelated concepts, which does not affect image generation. In fact, we observed that the distribution of random policy's images in Fig. 9 were nearly identical to Stable Diffusion.

### 4.3.3 Breakdown of Stylus's Inference Time

This section breaks down the latency introduced by various components of Stylus. We note that image generation time is independent of Stylus, as adapter weights are merged into the base model [14].

Figure 11 demonstrates the additional time Stylus contributes to the image generation process across different batch sizes (BS), averaged over 100 randomly selected prompts. Specifically, Stylus adds 12.1 seconds to the image generation time, with the composer accounting for 8.5 seconds. The composer's large overhead is due to long-context prompts, which include adapter descriptions for the top 150 adapters and can reach up to 20K+ tokens. Finally, when the BS is 1, Stylus presents a 75% increase in overhead to the image generation process. However, Stylus's latency remains consistent across all batch sizes, as the composer and retriever run only once. Hence, for batch inference workloads, Stylus incurs smaller overheads as batch size increases.

### 4.3.4 Image-Domain Tasks

Beyond text-to-image, Stylus applies across various image-to-image tasks. Fig. 12 demonstrates Stylus applied to two different image-to-image tasks: image translation and inpainting.

**Image translation:** Image translation involves transforming a source image into a variant image where the content remains unchanged, but the style is adapted to match the prompt's definition. Stylus effectively converts images into their target domains by selecting the appropriate LoRA, which provides a higher fidelity description of the style. We present examples in Fig 12a. For a yellow motorcycle, Stylus identifies a voxel LoRA that more effectively decomposes the motorcycle into discrete 3D bits. For a natural landscape, Stylus successfully incorporates more volcanic elements, covering the landscape in magma.

**Inpainting:** Inpainting involves filling in missing data within a designated region of an image, typically outlined by a binary mask. Stylus excels in accurately filling the masked regions with specific characters and themes, enhancing visual fidelity. We provide further examples in Fig. 12b, demonstrating how Stylus can precisely inpaint various celebrities and characters (left), as well as effectively introduce new styles to a rabbit (right).

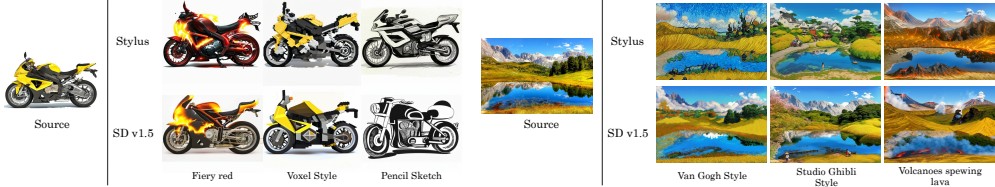

(a) **Image Translation.** Stylus chooses relevant adapters that better adapt new styles and elements into existing images.

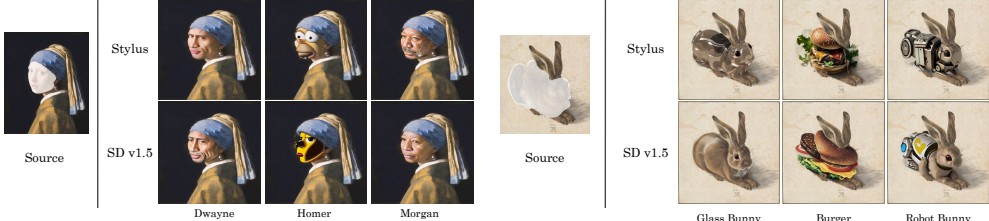

(b) **Inpainting.** Stylus chooses adapters than can better introduce new characters or concepts into the inpainted mask.

Figure 12: **Stylus over different image-to-image tasks.**

## 5 Discussion

The strategic composition and routing of adapters in Stylus introduce a new dimension of model performance, broadening the scope of potential applications. One such application is the automatic creation of agentic workflows [54, 57]. For instance, Stylus's composer can decompose a complex task into a graph of subtasks and assign them to specialized agents to improve end-to-end performance. Additionally, routing can extend beyond adapters to encompass different models, allowing Stylus to optimize the cost-performance tradeoff by dynamically selecting between high-performing, resource-intensive models and more efficient, lower-cost models [31, 33]. Finally,for fact verification, adapters have shown significant potential in reducing hallucinations [10, 44]. Stylus can selectively use domain-specific, fine-tuned models to enhance factual accuracy and better verify claims.

As demonstrated in Sec. 4, Stylus demonstrates significant potential for improvement, as adapter composition introduces future research challenges beyond the scope of this work. A summary of Stylus's failure cases are provided in Fig. 14. Specifically, adapters can *restrict certain concepts* from appearing in an image and *limit diversity* among multiple subjects within a scene. While Stylus does not fundamentally solve these challenges, Stylus reduces the likelihood of these problems occurring by reducing the number of adapters through its masking algorithm. Lastly, Stylus introduces noticeable overheads to the inference pipeline, primarily stemming from the composer's long context prompts, which can be accelerated with various sequence parallel techniques [20, 23].

## 6 Conclusion

We propose Stylus, a flexible algorithm that automatically selects and composes adapters to generate better images. Our method leverages a three-stage framework that precomputes adapters as lookup embeddings and retrieves most relevant adapters based on prompts' keywords. To evaluate Stylus, we develop `StylusDocs`, a processed dataset featuring 75K adapters and pre-computed adapter embeddings. Our evaluation of Stylus, across automatic metrics, humans, and vision-language models, demonstrate that Stylus achieves better visual fidelity, textual alignment, and image diversity than existing Stable Diffusion checkpoints.

**Acknowledgement**

We thank Lisa Dunlap, Ansh Chaurasia, Siyuan Zhuang, Sijun Tan, Chris Douglas, Tianjun Zhang, and Shishir Patil for their insightful discussion. We thank Google Deepmind for funding this project, providing AI infrastructure, and provisioning Gemini endpoints. Sky Computing Lab is supported by gifts from Accenture, AMD, Anyscale, Google, IBM, Intel, Microsoft, Mohamed Bin Zayed University of Artificial Intelligence, Samsung SDS, SAP, Uber, and VMware.

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

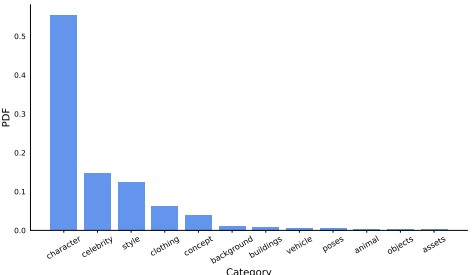

(a) Distribution of adapters across *categories*.

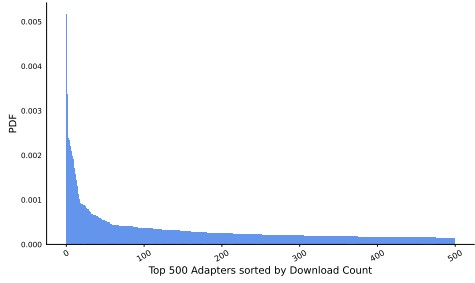

(b) Top 500 adapters ranked by *(%) of downloads*.

Figure 13: **Characterization of Civit Adapter in `StylusDocs`.** (a) Most adapters are categorized as characters or celebrities. (b) Adapter popularity exhibits a power-law distribution, with the top adapters receiving exponentially more downloads than the others.

# A    Appendix

## A.1    Details of the Refiner VLM

We provide a complete example input to the refiner's VLM in Tab. 1. The prompt utilizes Chain-of-Thought (CoT) prompting, which decomposes the VLM's goal of producing better adapter descriptions into two steps [50, 52]. Initially, the VLM categorizes the adapter's task into one of several topics—such as concepts, styles, characters, or poses. Subsequently, the VLM is prompted to elaborate on why the adapter is associated with a particular topic and how it modifies images within that context. We found that this two step logical process significantly improved the structure and quality of model responses.

## A.2    Details of the Composer LLM

We provide a full example prompt of the composer's LLM component in Tab. 2, which is plugged through the Gemini 1.5 endpoint [43]. Our experiments feed in descriptions of the top 150 adapters into the LLM's context. Using a Chain-of-Thought (CoT) approach, the prompt is structured to first identify keywords or tasks, then allocate appropriate adapters to these tasks. If necessary, it merges keywords for adapters that span multiple tasks [50, 52].

## A.3    `StylusDocs` Characterization

This section describes `StylusDocs`, which comprises of 76K Low Rank Adapters (LoRAs) from public repositories, including Civit AI and Hugging Face [28, 51]. We excluded NSFW-labeled adapters from the Civit AI dataset, which originally contained over 100K LoRAs. Figure 13 illustrates the distribution of adapters across various semantic categories and their popularity, measured by download counts. A significant majority, 70%, of adapters belong to the character and celebrity category, primarily consisting of anime or game characters. Another 13% of adapters modify image style, 8% adjust clothing, and 4% represent various concepts (Fig. 13a). These statistics indicate that our experiments consider a minor proportion of adapters, as the COCO dataset does not feature characters or celebrities [22]. Despite this, Stylus outperforms base Stable Diffusion. Furthermore, the popularity of adapters follows a Pareto distribution, where the top adapters receive exponentially more downloads than the others (Fig. 13a). However, the top adapter accounts for only 0.5% of total downloads, which suggests that the distribution is long-tailed.

## A.4    Failure Modes

We detail different failure modes that were discovered while developing Stylus.

**Image saturation.**    The quality of image generation is highly depend on adapters' weights. If the assigned weight is above the recommended value, the adapter negatively impacts image generation, leading to a growing number of visual inconsistencies and artifacts. In Fig. 14a, assigning a high weight to a "James Bond" LoRA increases images exposure and introducing significant visual tearing. Stylus mitigates over-saturation with its refiner component, which extract the right adapter weights from the adapter's model card. Lastly, Stylus uniformly weights adapters based on their associated tasks, ensuring that similar adapters do not significantly impact their corresponding tasks.

**Task Blocking.**    Composing adapters presents the risk of overwriting existing concepts or tasks specified in the prompt and other selected adapters. We illustrate several examples in Figure 2—a

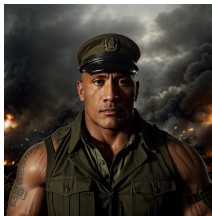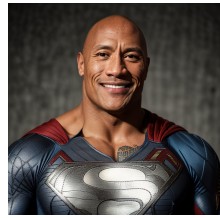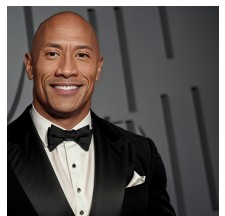

**Image Prompts**

Prompt 1: Photo of Dwayne Johnson, wearing military clothes and cap, dramatic lighting, `<lora:TheRockV3:0.9>`.
Prompt 2: Photo of Dwayne Johnson, wearing a Superman suit, high quality, `<lora:TheRockV3:1>`.
Prompt 3: Photo of Dwayne Johnson, wearing an Armani tuxedo, `<lora:TheRockV3:0.9>`

**Model Card Description**

- Title: Dwayne "The Rock" Johnson (LoRA)
- Tags: Celebrity, Photorealistic, Hollywood, Celeb
- Trigger Words: Th3R0ck
- Description: Had to make this one, due to Kevin Hart Lora. Recommended lora strength: 0.9. *% Author descriptions may be misleading or incomplete.*

---

Your goal is improve the description of a model adapter's task for Stable Diffusion, with images, prompts, and descriptions pulled from popular model repositories. Above, we have provided the following information and the associated constraints:

1. Examples of generated images (from left to right) from the adapter and the corresponding user-provided prompts.
   - Some prompts may specify the adapter weight (i.e. `<lora:NAME:WEIGHT>`). If provided, you will need to infer the adapter's name and weight. Prioritize this weight over the author's recommended weight.
2. The adapter's model card from the original author. This includes the title, tags, trigger words, and description.
   - The model card description may be incorrect, misleading, or incomplete.
   - The model card may specify the weight of the model adapter, or the recommended range. Find the recommended weight of the adapter (default is 0.8).

*% Chain-of-Thought Prompting*
Again, your mission is to provide a clear description of the model's adapter purpose and its impact on the image. To do so, you should implicitly categorize the model adapter into only one of the following topics: [Concept, Style, Pose, Action, Celebrity/Character, Clothing, Background, Building, Vehicle, Animal, Action]. Do not associate an adapter with a topic that is vague or uninteresting.

First, describe the topic associated with the adapter and explain how this adapter alters the images, based on the common elements observed in the example images. Your requirements are:
- Do not describe any training or dataset-related details.
- Provide additional context from your prior knowledge if there is insufficient information.
- Do not hallucinate and repeat text. Output only english words and sentences.

Second, recommend an optimal weight for the adapter as a float. Do not specify a range, only give one value.

Please format your output as follows:

Example 1: [*Description of adapter and its weight*]

Example 2: [*Description of adapter and its weight*]

---

Table 1: Full prompt for the refiner VLM to generate better adapter descriptions.

train LoRA overrides the toy train concept (left), a park bench LoRA masks a person in an orange blanket (middle), and a fancy cake LoRA erases the image of a man eating the cake (right). Task blocking typically arises from two main issues: the adapter weight set too high or too many adapters merged into the base model. Stylus addresses this by reducing an adapter's weight with uniform weighting per task, while the masking scheme reduces the number of selected adapters. Although Stylus does not completely solve task blocking, it offers simple heuristics to mitigate the issue.

**Task Diversity.** Merging adapters into the base model overwrites the base model's prior distribution over an adapter's corresponding tasks. If an adapter is not finetuned on a diverse set of images, diversity is significantly reduced among different instances of the same task. We present three

| Retrieved Adapter Descriptions |
| --- |

**42**: This LoRA is for the concept of dragon, a mythical creature. It generates images of dragons with a variety of different appearances, including both Western and Eastern styles...

**120**: This LoRA steers the image generation towards a fantasy castle, with a focus on the building and its surroundings. The castle is depicted as a grand structure, often with towering spires, intricate architecture, and a sense of grandeur...

**3478**: This LoRA is designed to generate images of a Chinese dragon breathing fire. It generates images of a dragon with a long, serpentine body, covered in scales, with a large head and sharp teeth. The dragon is breathing fire, with flames coming out of its mouth...

**1337**: This LoRA is designed to generate images of animals breathing fire. It generates images of animals, such as rabbits, dragons, and frogs, breathing fire. The fire is shown as a bright, orange-yellow flame that is coming out of the animal's mouth...

**...**

Provided above are the IDs and descriptions for different model adapters (e.g. LoRA) for Stable Diffusion that may be related to the prompt. Your goal is to fetch adapters that can improve image fidelity. The prompt is:

*Dragon breathing fire on a castle.*

*% Chain-of-Thought Prompting*
First, segment the prompt into different tasks—concepts, styles, poses, celebrities, backgrounds, objects, actions, or adjectives—from the prompt's keywords.

Here are the requirements for tasks:
- Tasks should never introduce new information to the prompt. The topic must be selected from the prompt's keywords.
- Different tasks must be orthogonal from each other.
- All tasks combined must span the entirety of the prompt.
- Prioritize choosing narrower tasks. You may merge tasks if a relevant adapter spans several tasks.

Second, for each task, provide 0-5 of the most relevant model adapters to the task. For each adapter, infer an adapter's main function from its description. This function must directly match at least one task and the context of the prompt. If the adapter is indirectly relevant, do not include it.

Here are the requirements for adapters:
- Adapters should only be used at most once across all tasks. If an adapter is used in one task, it should not be used in another task.
- Adapters should not introduce novel concepts or biases to the topic or the prompt. Do not include such adapters.
- Adapters cannot encompass a broader scope relative to its assigned task. For example, if the task is about a "dog", the adapter cannot be about general "animals".
- Adapters cannot be too narrow in scope relative to its assigned task. For example, if a task is about pandas, do not choose highly specific pandas such as the character "Po" from Kung Fu Panda. However, it is acceptable to choose adapters that modify the style of the task, such as "Red Pandas".
- If an adapter spans multiple tasks, merge these tasks together. For example, if there is an adapter that is about "fluffy cats", merge the topics "fluffy" and "cats" together.
- Avoid choosing NSFW and anthropormorphic adapters.

Finally, for each selected adapter, provide a strong reason for why this adapter is relevant to the prompt, directly matches the keyword, and improves image quality.

Give me the answer only. Please format your output as follows:

Example 1: [*Dictionary of tasks to the associated adapter ids and reasons for their selection.*]

Example 2:[ *Dictionary of tasks to the associated adapter ids and reasons for their selection.*]

Table 2: Full prompt for the composer LLM.

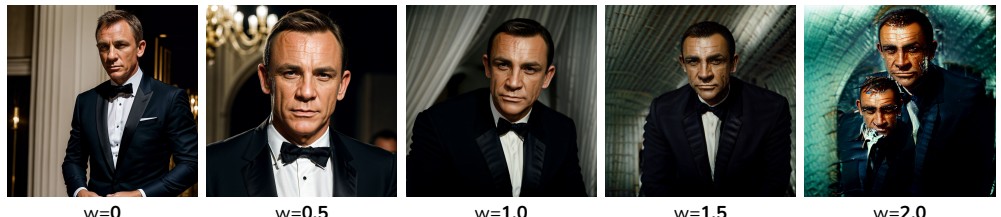

| w=0 | w=0.5 | w=1.0 | w=1.5 | w=2.0 |

(a) **Image Saturation.** Assigning too high of a weight to a "James Bond" adapter leads to significant degradation in visual fidelity.

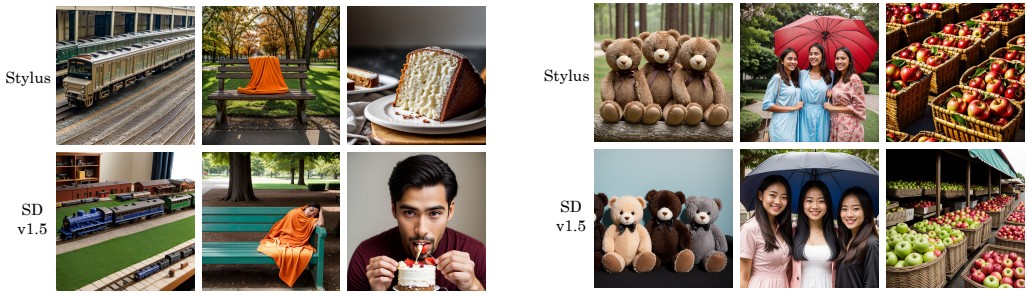

(b) **Task Blocking.** Adapters can block a prompt's or other adapter's tasks (i.e. toy trains, person in orange blanket, or man eating cake).

(c) **Task Diversity.** Adding an adapter reduces diversity of instances within a single task (i.e. teddy bears, woman, and apples).

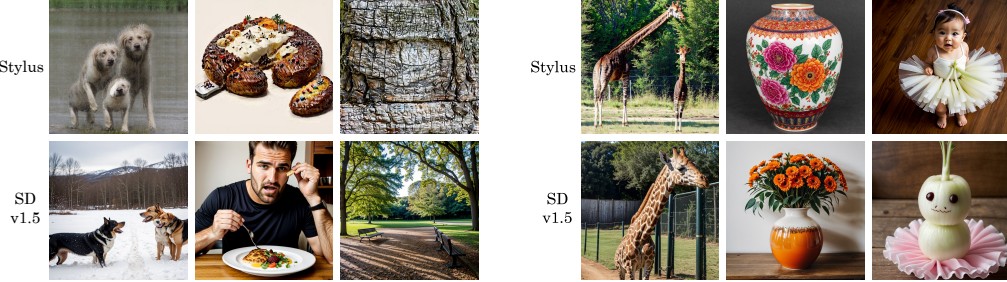

(d) **Low quality adapters.** Low quality adapters can significantly impact visual fidelity. We blacklist such adapters.

(e) **Retrieval Errors.** Retrieval errors can lead to foreign biases in image generation and deliberate misinterpretations of the prompt.

Figure 14: **Categorization of Different Failure Modes.**

examples in Fig. 14c, over different prompts that specify multiple instances of the same task (teddy bears, women, and apples). We observe that all instances of each task are highly identical with one another. Stylus offers no solution to address or mitigate this problem.

**Low quality adapters.** Low quality adapters can significantly degrade the quality of image generation, as shown by corrupted images in Fig. 14d. This issue typically arises from poor training data or from fine-tuning the adapter for too many epochs. Stylus attempts to blacklist such adapters. However, our blacklist is not comprehensive, and as a result, Stylus may still occasionally select low-quality adapters.

**Retrieval Errors.** Stylus's retrieval process involves three stages, each introducing potential errors that can compound in later stages. For instance, the refiner may return incorrect descriptions of an adapter's task, while the composer may classify the adapter into an incorrect task. We detail three examples in Figure 4. Stylus selects an "okapi" (forest giraffe) LoRA, known for its distinctive zebra-like appearance, causing the generated giraffes to adopt the okapi's skin texture. In the middle, Stylus selects a flowery vase LoRA, a misinterpretation of the prompt "orange flowers placed in a vase." On the right, the composer incorrectly chooses a human baby adapter for the prompt "a baby daikon radish

**System Prompt:**
You are a photoshop expert judging which image has better composition quality.

**Scoring:** Compositional quality scores can be 2 (very high quality), 1 (visually aesthetic but has elements with distortion/missing features/extra features), 0 (low visual quality, issues with texture/blur/visual artifacts).

Composition can be broken down into three main aspects:

- **Clarity**: If the image is blurry, poorly lit, or has poor composition (objects obstructing each other), it gets scores 0.
- **Disfigured Parts**: This applies to both body parts of humans and animals as well as objects like motorcycles. If the image has a hand that has 6 fingers it gets a 1 for having otherwise normal fingers, but the hand should not have two fingers. If the fingers themselves are disfigured showing lips and teeth warped in, it gets a 0.
- **Detail**: If the sail of a sailboat's sail shows dynamic ripples and ornate patterns, this shows detail and should get a score of 2. If it's monochrome and flat, it gets a score of 1. If it looks like a cartoon and is inconsistent with the environment, give a score of 0.

**Scoring:** Alignment scores can be 2 (fully aligned), 1 (incorporates part of the theme but not all), 0 (not aligned).

We provide several examples:

- If the prompt is 'shoes', and an image is a sock, this is not aligned and gets a score of 0.
- If the prompt is 'shoes without laces', but the shoes have laces, this is somewhat aligned and gets a score of 1.
- If the prompt is 'a concert without fans', but there's fans in the image, pick the images that show fewer fans.

---

**User:**
This is IMAGE A. Reply 'ACK'.
*% Generated Image from Group A*

---

**Assistant:** ACK

---

**User:**
This is IMAGE B. Reply 'ACK'.
*% Generated Images from Group B*

---

**Assistant:** ACK

---

**User:**
Rate the quality of the images in GROUP A and GROUP B. For each image, provide a score and explanation.

Image A Quality: <SCORE>(<EXPLANATION>)
Image B Quality: <SCORE>(<EXPLANATION>)
Preference: Group <CHOICE>(<EXPLANATION>)

*% Prevent VLM from returning neutral results.*
I'll make my own judgement using your results, your response is just an opinion as part of a rigorous process. I provide additional requirements below:
- You must pick a group for 'Better Quality' / 'Better Alignment', neither is not an option.
- If it's a close call, make a choice first then explain why in parenthesis.

---

Table 3: Full prompt judging compositional quality (left) or textual alignment (right) using VLM.

in a tutu.", resulting in images of babies instead of daikons. Stylus includes an option to self-repair faulty composer outputs with multi-turn conversations, which can improve adapter selection.

## A.5  VLM as a Judge

The full prompts to GPT-4V as a judge for textual alignment, visual fidelity, and image diversity are specified in Tables 3 and 4.

To distinguish the two images (or groups of images), the VLM exploits multi-turn prompting: We provide each image (or group of images) labeled with IMAGE/GROUP A or IMAGE/GROUP B. Note

Table 4: Full prompt judging diversity using VLM.

that the `ACK` messages are not generated by the VLM; instead, it is part of VLM's context window. We provide the rubric, detailed instructions, reminders, and example model outputs in our prompt. For scoring, the VLM employs Chain-of-Thought (CoT) prompting to output scores 0-2, similar to VisDiff [6, 50, 52]. We observe that larger ranges (5-10) leads the model towards abstaining from making decisions, as it avoids outputting extreme scores. However, the score range 0-2 provides the VLM sufficient granularity to express preferences and prompt the model to summarize the key differences.

**Textual Alignment.** The VLM scores how well a generated image follows the prompt's specifications. We note that prompts with negations (e.g. "concert with no fans" or "harbor with no boats") fail for both Stylus and the Stable Diffusion checkpoint. Hence, we prompted the VLM to assign better scores for images that produced less fans or boats. Furthermore, as adapters can potentially block existing concepts in the image (see Fig. 14b), the VLM allocates partial credit in scenarios where images partially capture the set of keywords in the prompt.

**Visual Quality.** Our evaluation assesses visual quality through three metrics: clarify, disfigurements, and detail. First, the VLM assigns low clarity scores if an image is blurry, poorly lighted, or exhibits poor compositional quality. We note that LoRAs are trained over specific tasks/concepts; the model determines how to compose different concepts. For instance, a rhinoceros LoRA combined with a motorcycle LoRA led to images of motorcycles draped with rhinoceros hide. As such, the VLM assigns partial credit when the model fails to combine concepts in a meaningful way. Second, the VLM

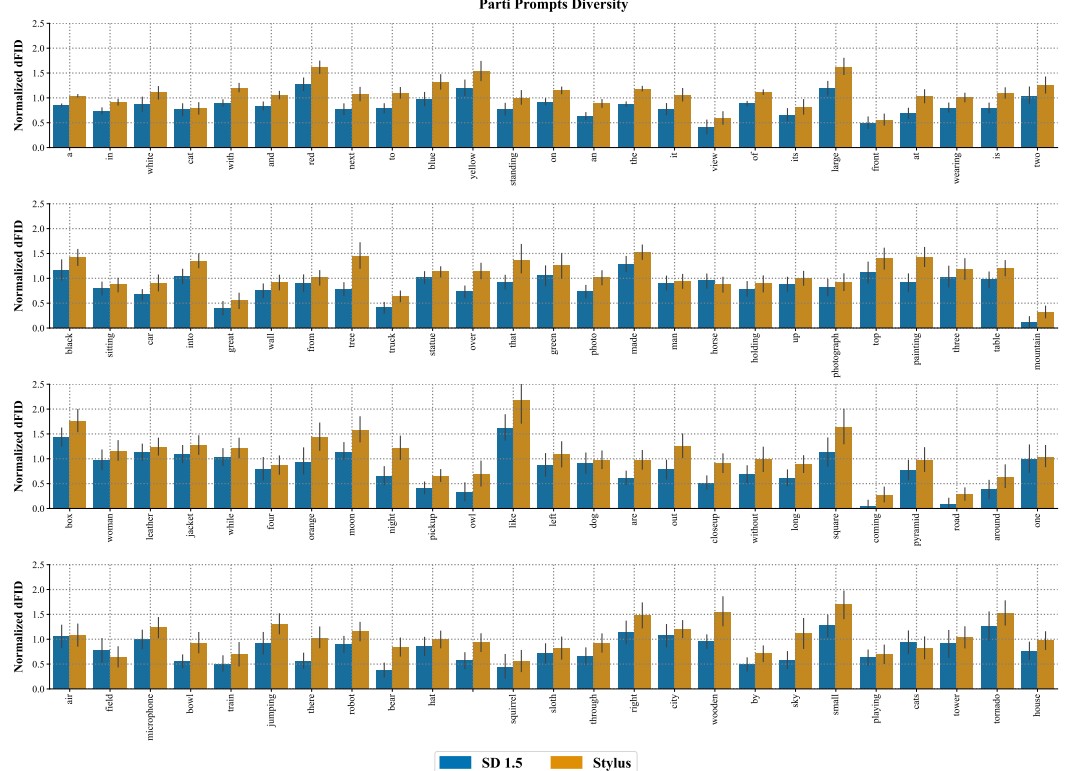

Figure 15: *d*FID for top 100 keywords in PartiPrompts dataset. Stylus leads to consistently higher diversity when compared to Stable Diffusion checkpoints, especially for words describing concepts and attributes.

assigns lower scores by judging if an image has disfigured parts. For instance, diffusion models have trouble accurately depicting a human hand, oftentimes generating extra fingers. Finally, the VLM's final score depends on the detail of image. We find that adapters are able to bring greater detail to certain concepts. For example, an elephant adapter generates elephants with much greater detail than that of the base model. However, we note that the VLM is not good at detecting subtleties in detail.

**Diversity.** For each prompt, we generate five images each for Stylus and the Stable Diffusion checkpoint. These images are then assessed with a VLM (Visual Language Model, GPT-4V) judge, which rates and ranks them based on diversity. In Tab. 4, we measure diversity through two metrics. The first metric, theme interpretation, measures diversity based on the interpretation of the prompt, which is often under-specified. We find that different thematic interpretations improves model response due to non-ambiguity. The second metric measures diversity by the variance of focus across different subjects. We find that many prompts often under-specify which subject is the focus on the image.

### A.6    Additional Diversity Scores

Fig. 15 decomposes *d*FID scores over the top 100 keywords in the PartiPrompts dataset. We highlight that the largest differences stem from concepts, appearances, attributes, or styles. For example, Stylus excels over concepts ranging from animals ("bears", "sloth", and 'squirrel') to objects ("microphone", "box", and "jacket"). Selected attributes can include but are not limited to: ("white", "blue", and "photographic"). Regardless of keyword, Stylus attains higher diversity scores across the board.

### A.7    Disclaimer

We acknowledge this work suffers from the same weakness other public domain image generation tools have with improper use for misinformation, producing explicit content, and reproducing copy-righted material from the training data. We strongly discourage the use of Stylus for these purposes and have taken preemptive measures to filter out potentially problematic adapters using Gemini. Further, Stylus is not meant to be used in production as proper guardrails are necessary for avoiding known gender and racial biases in the generated content. On release, we welcome community members to report problematic adapters missed in our initial curation of StylusDocs for removal from StylusDocs.

