# OpenReview forum: "Stylus: Automatic Adapter Selection for Diffusion Models"
_NeurIPS.cc/2024/Conference — NeurIPS 2024 oral_

### Official Review · Reviewer_wGTi · 2024-07-10

**Soundness:** 3
**Presentation:** 2
**Contribution:** 3
**Rating:** 7
**Confidence:** 4

**Summary:**

This work presents a novel system, Stylus, designed to enhance the efficiency and effectiveness of generating high-quality images using diffusion models like Stable Diffusion. The key challenge addressed by this work is the automatic selection and composition of relevant fine-tuned adapters from a vast pool of over 100,000 available adapters, which are often poorly documented and highly customized. This work advances the field by providing a robust and automated solution by leveraging the vast number of available adapters.

**Strengths:**

- The creation of StylusDocs, a curated dataset with 75K adapters and pre-computed embeddings, adds substantial value to provide a rich resource for further experimentation and development.
- Comprehensive Evaluation: The paper provides a thorough evaluation of Stylus across multiple metrics (CLIP, FID) and diverse datasets (Microsoft COCO, PartiPrompts). This robust evaluation framework enhances the credibility of the claimed performance improvements.
- Open Source and Reproducibility: By planning to release Stylus and StylusDocs as open-source resources, the authors contribute to the transparency and reproducibility of their research. This aligns well with the community’s values and encourages further developments based on their work.

**Weaknesses:**

- Unclear Motivation for the Method: In the refiner step, the paper does not clearly explain why Gemini Ultra is trusted to generate better adapter descriptions. If other multimodal language models (MLLMs) were used, how would the results differ?
- Incomplete Ablation Study: The ablation study is not comprehensive as it does not include an ablation of the refiner component. Understanding the impact of the refiner step on the overall performance of Stylus is crucial.
- Quality Assurance of Adapter Descriptions: The paper does not provide sufficient details on how the quality of the adapter descriptions generated by the refiner is ensured. It is unclear whether any validation or verification steps were taken to confirm the accuracy and reliability of these descriptions.
- Insufficient Description of Masking Process: The description of the masking process is not detailed enough. Specifically, the meaning of α_j in Equation 2 and the function Mask() are not adequately explained. Additionally, the masking process is not reflected in Figure 2, which outlines the Stylus algorithm.

**Questions:**

- Motivation of the selected MLLM
- Additional ablation study
- Check the quality of adapter descriptions
- Details on masking process

**Limitations:**

Maybe potential bias and fairness issues will related with this work

---

> ### Author Rebuttal · Authors · 2024-08-05
>
> We appreciate the thoughtful review and the constructive feedback! We hope the following clarifications address all the questions raised.
>
> **Quality Assurance for Adapters.** We have taken careful preemptive measures to filter out problematic adapters and have outlined in Appendix A.7 a continual curation process, where we invite community members to report any adapters missed in our initial curation. Below, we highlight some key preventative measures for quality assurance and refer readers to Appendix A.7 and A.8 for a full discussion on safety and reliability of Stylus.
>
> - **Safety.** Although LLM safety remains an actively evolving research area [1], StylusDocs employs a multi-stage filtering pipeline to identify and exclude problematic adapters. First, all explicit adapters tagged by Civit.ai are excluded from StylusDocs (Sec 3.2). Second, we use filters from Google’s VertexAI API to reject unsafe adapters based on LLM-generated descriptions. This especially catches problematic adapters that have innocuous original descriptions (Sec. A.7).
> - **Accuracy/Reliability.** We note that, without access to the original fine-tuning dataset, we cannot guarantee the descriptions are completely error-free. However, we manually inspect commonly selected adapters in StylusDocs, blacklisting adapters that produce low quality images or cases where the StylusDocs description is inconsistent with observed adapter behavior (Sec A.4). Furthermore, authors are incentivized to oversell the model’s abilities. As such, Stylus’s refiner takes in example images in Civit.ai’s model card as a grounding mechanism to generate more accurate descriptions (Sec A.1).
>
> Even with the safety measures we’ve taken, we acknowledge Stylus has the same risks of misuse as other public domain image generation tools including improper use for misinformation, producing explicit content, and reproducing copy-righted material from the training data. We emphasize the research prototype is not meant to be used in production without further application informed guardrails.
>
> **Refiner Ablation.** In the following table containing CLIP and FID scores, we ablate Stylus's refiner:
>
> | Baselines (CFG=6)        | CLIP: ↑ is better     | FID: ↓ is better      |
> |--------------------------|-----------------------|-----------------------|
> | SD v1.5                  | 27.22                 | 23.96                 |
> | No-Refiner               | 24.91 (-2.31)         | 24.26 (+0.3)          |
> | Gemini-Ultra Refiner     | 27.25 (+0.03)         | 22.05 (-1.91)         |
> | **GPT-4o Refiner**                  | **28.04 (+0.82)**     | **21.96 (-2.00)**     |
>
> The baselines are:
> - SD v1.5 - The base Stable Diffusion model with the RealisticVision checkpoint.
> - No-Refiner: Use base author-provided descriptions from Civit.ai or Huggingface.
> - Gemini-Ultra Refiner: Use Gemini-Ultra as the Refiner’s VLM to generate better adapter descriptions. This is the version of Stylus presented throughout our paper.
> - GPT-4o Refiner: Use GPT-4o as the Refiner’s VLM to generate better adapter descriptions.
>
> These results show that a refiner is indeed important for textual alignment with the prompt. In fact, without a refiner VLM, Stylus performs poorly and chooses adapters that the composer thinks are aligned with the prompt but are not in practice. As a result, Stylus chooses adapters that hurt textual alignment (CLIP) and image quality (FID) and performs worse than base SD.
>
> Furthermore, GPT-4o baseline performs much better than Gemini-Ultra, showing that better refiner descriptions can better aid the composer in selecting the right adapters. This also suggests that the Stylus’s performance is independent of Gemini-specific capabilities and benefits from further improved visual-language reasoning capabilities.
>
> **Choice of Gemini for Refiner.** We chose the Gemini class of models since it has mature safety guardrailing. Specifically, Google’s VertexAI API provides stringent safety settings to block explicit content for the input prompt. Safety filters helped us filter out around 30% of original adapters that were tagged as non-explicit by Civit.ai.
>
> **Masking.** The composer decomposes a prompt into tasks and assigns highly-aligned adapters per task. Next, a subset of candidate adapters are selected via masking for image generation. For each task, a mask either selects A) just one of the task’s adapters, B) all of the task’s adapters, C) or none of the adapters. To get the final selection of adapters, we randomly sample a mask for each task and merge the identified adapters to the original base model.
>
> Regarding merging adapters, each adapter weights are first multiplied by the refiner’s recommended adapter weight, α_j (Eqn. 2). For example, [Food Elegant Style LoRA](https://civitai.com/models/127450?modelVersionId=139441) recommends α=0.7 weight. Finally, to merge adapters into the base model, adapters weights are *averaged* per task and then *summed* across tasks.
>
> We appreciate your feedback and believe these clarifications should address your concerns. We are open to further discussions to improve the paper. Thank you once again for your valuable insights!
>
> [1] Hendrycks, Dan, et al. "Unsolved Problems in ML Safety." arXiv, 29 Sep. 2021, arXiv:2109.13916.

---

> > ### Comment · Reviewer_wGTi · 2024-08-08
> >
> > Thank you for your response. Most of my concerns have been addressed. However, I suggest adding more details about the masking process to the manuscript for clarity. I'm pleased to raise my score.

---

> > > ### Author Response · Authors · 2024-08-08
> > >
> > > Thanks! We will definitely revise the description of the masking process in the manuscript to reflect the clarification we provided in the rebuttal.

---

### Official Review · Reviewer_N5AD · 2024-07-12

**Soundness:** 4
**Presentation:** 4
**Contribution:** 4
**Rating:** 9
**Confidence:** 4

**Summary:**

The paper addresses the challenge of selecting and composing relevant adapters for generating high-fidelity images with diffusion models. Stylus introduces a three-stage process: refining adapter descriptions, retrieving relevant adapters based on user prompts, and composing them to create the final image. The paper highlights the development of StylusDocs, a dataset featuring 75K adapters with pre-computed embeddings. Evaluation results show that Stylus outperforms baseline models, achieving higher visual fidelity, textual alignment, and image diversity. The system is efficient and suitable for various image-to-image tasks, including translation and inpainting, demonstrating its versatility and effectiveness in improving image generation.

**Strengths:**

- This is a great paper. Original, high quality, clear, and significant.
- The use of adapters and PEFT will/should continue to increase in the future. The authors present a scalable method to improve text-to-image generation.

**Weaknesses:**

None. While adapters have been used in the past to improve image generation, this paper provides a much more coherent strategy to integrate them into VLMs.

**Questions:**

None.

**Limitations:**

None. Great work, Authors!

---

> ### Author Rebuttal · Authors · 2024-08-05
>
> We appreciate the thoughtful and enthusiastic review!
>
> We agree the incorporation of adapters/PEFTs [1] will continue to increase and that automatic adapter selection will be critical for managing and navigating the growing ecosystem of fine-tuned models. Stylus demonstrates improved diversity and visual quality evaluated quantitatively with automated metrics (CLIP/FID scores) as well as qualitatively via human evaluation and VLM as a judge.
>
> Stylus provides a coherent strategy for composing and routing adapters for VLMs. Our retriever ablation (Sec. 4.3.1, Tab. 1) shows that naively selecting adapters (i.e. RAG) can lead to worse performance than the base SD checkpoint. Strategically composing and routing adapters unlocks a dimension of model performance previously underexplored.
>
> We are excited to see LLMs used for dynamically selecting among models in related domains, including but not limited to:
> - Automatic construction of agentic workflows/graphs. This includes using Stylus to decompose the task/prompt into a graph of subtasks and identifying which agent, among an ecosystem of agents, is best suited for each subtask.
> - Routing between different base models from different providers to optimize the cost-performance tradeoff.
> - Given a user prompt that requires composing multiple tools/functions, Stylus can identify, retrieve, and then compose the right sets of tools and functions for the LLM to invoke.
> - Domain-specific fine-tuning is an emerging approach to reduce hallucination [2]. Stylus can select the right domain fine-tuned model to maximize factuality.
>
> We are open to further discussions to improve the paper. Thank you once again for your valuable insights!
>
> [1] Hugging Face Team. "Parameter-Efficient Fine-Tuning (PEFT) with Hugging Face." GitHub, 2023, https://github.com/huggingface/peft.
>
> [2] Tian, Katherine, et al. "Fine-tuning Language Models for Factuality." arXiv, 2023, arxiv.org/abs/2311.08401.

---

### Official Review · Reviewer_rCTt · 2024-07-14

**Soundness:** 3
**Presentation:** 3
**Contribution:** 3
**Rating:** 7
**Confidence:** 3

**Summary:**

The paper proposes Stylus, an approach for automatically selecting and combining fine-tuned adapters on particular tasks to improve the quality of image generation given a prompt. To evaluate Stylus, the paper introduces StylusDocs, a curated dataset containing 75K adapters with pre-computed adapter embeddings. Both the qualitative and quantitative results show that the proposed method outperforms Stable Diffusion and other retrieval methods.

**Strengths:**

- The paper explores an interesting topic of automatically selecting and combining fine-tuned adapters on particular tasks to improve the quality of image generation.
- Both the qualitative and quantitative results are proposing. The proposed methods method improves over other method for both human evaluation and automatic benchmarks.

**Weaknesses:**

- Some details about the proposed method are missing, making it hard to reproduce. In particular, sections 3.3 and 3.4 about the composer and the masking are not very clear. How are the adapters selected? How is masking applied?

**Questions:**

No questions beyond the one in the weaknesses.

**Limitations:**

The paper does not have any obvious limitations that were not discussed.

---

> ### Author Rebuttal · Authors · 2024-08-05
>
> We appreciate the thoughtful and enthusiastic review!
>
> **Reproducibility.** We plan to release Stylus and StylusDocs as open-source resources to ensure transparency and reproducibility.
>
> **Adapter selection and masking.** The composer decomposes the prompt into tasks and maps highly relevant adapters to each task. For more details on the Chain-of-Thought prompt [1] used for Stylus’s composer, refer to Table 2.
>
> Next, a subset of candidate adapters are selected via masking for image generation. For each task, a mask either selects A) just one of the task’s adapters, B) all of the task’s adapters, C) or none of the adapters. To get the final selection of adapters, we randomly sample a mask for each task and merge the identified adapters with the original base model.
>
> Regarding the merging step, each adapter weights are first multiplied by the refiner’s recommended adapter weight, α_j (Eqn. 2). For example, [Food elegant style LoRA]( https://civitai.com/models/127450?modelVersionId=139441) recommends α=0.7 weight. Finally, to merge adapters into the base model, adapters weights are *averaged* per task and then *summed* across tasks.
>
> We appreciate your feedback and believe these clarifications should address your concerns. We are open to further discussions to improve the paper. Thank you once again for your valuable insights!
>
> [1] Wei, Jason, et al. “Chain-of-Thought Prompting Elicits Reasoning in Large Language Models.” arXiv.Org, 28 Jan. 2022, https://arxiv.org/abs/2201.11903v6.

---

> > ### Comment · Reviewer_rCTt · 2024-08-13
> >
> > Thank you for your response. My concerns have been addressed and I will keep my initial score.

---

> > > ### Author Response · Authors · 2024-08-13
> > >
> > > Thanks! We appreciate your enthusiastic review and constructive feedback!

---

### Official Review · Reviewer_QQJ7 · 2024-07-17

**Soundness:** 3
**Presentation:** 3
**Contribution:** 3
**Rating:** 7
**Confidence:** 4

**Summary:**

This paper works on post-training optimization for image generation with stable diffusion models. They proposed three stages, refiner, retriever and composer, to personalize a SD model for the prompt and thus to generate the perfect images. The experimental result indicate the potential of the proposed method.

**Strengths:**

1. The motivation makes sense to me and the idea is interesting. Previously, we usually explore prompt engineering to generate a good image, but this paper investigates the adapters and to finalize a suitable model to generate  good image given a fixed prompt.
2. The post-training optimization consists of three stages, Refiner, Retriever and Composer, the design of the entire method is reasonable.
3. The experimental results demonstrate the proposed method is promising.

**Weaknesses:**

1. Efficiency. In the paper, the authors compares the Stylus with the typical SD checkpoint, then the efficiency of stylus is not comparable to SD in terms of memory, CPU, GPU resources.
2. Fairness. The paper claim advantages over typical SD in terms of diversity and quality. While it is not quite fair. The Stylus has a model personalization process (from the post-training optimization pipelines), while the SD is using a static model checkpoint.

**Questions:**

1. Does the description (or the optimized version) can well represent the adapter?
2. in Eq.2, why do you directly use betha=0.8? If the 2nd term's value is much bigger than W_base, how do you deal with it?

---

> ### Author Rebuttal · Authors · 2024-08-05
>
> We appreciate the thoughtful review and the constructive feedback! We hope the following clarifications and experiments address your questions.
>
> **Efficiency.**  We note that Stylus's efficiency for image generation in terms of CPU and GPU resources is near identical to the base Stable Diffusion (SD) model. Aside from merging adapters (e.g. LoRAs) into the base model, which is small, the inference computation remains the same. Additional CPU memory is required to store unmerged adapters, as evidenced by efficient LoRA serving systems such as SLoRA [1] and dLoRA [2].
>
> We emphasize that users today rely on manual search to identify a helpful subset of adapters. Stylus automates the process of determining the best set of adapters for a given user prompt, improving user efficiency. Furthermore, with recent releases of fast, high quality LLMs (i.e. GPT-4o-mini, Gemini 1.5 Flash), the composer’s latency will continue to reduce over time. We profiled GPT-4o-mini as the composer, which is over 3x faster than Gemini 1.5. This is a small overhead when compared to manual search over adapters.
>
> **Fairness.** Stylus serves as an automatic enhancement to SD, leveraging additional training and data represented by LoRAs. Our retriever ablation (Sec. 4.3.1, Tab. 1) shows that naively selecting adapters (i.e. RAG) can lead to worse performance than the typical SD checkpoint. As such, we include base SD’s CLIP and FID scores as a reference for comparing different approaches to selecting and composing adapters.
>
> **Adapter descriptions.** First, we note that, without access to the original fine-tuning dataset, we cannot guarantee the descriptions are completely error-free. However, we manually inspect commonly selected adapters in StylusDocs, blacklisting adapters that produce low quality images or cases where the StylusDocs description is inconsistent with observed adapter behavior (Sec A.4).  Furthermore, we observe that over 80% of adapters on model platforms (Civit.ai/Huggingface) lack sufficient descriptions. As such, Stylus’s refiner takes in example images from Civit.ai’s model card as a grounding mechanism to generate more detailed and accurate descriptions (Sec A.1).
>
> Furthermore, we add an additional refiner ablation that showcases that better adapter descriptions lead to performance gains. In the following table of CLIP/FID scores, we illustrate Stylus without and with refiner.
>
> | Baselines (CFG=6)        | CLIP: ↑ is better     | FID: ↓ is better      |
> |--------------------------|-----------------------|-----------------------|
> | SD v1.5                  | 27.22                 | 23.96                 |
> | No-Refiner               | 24.91 (-2.31)         | 24.26 (+0.3)          |
> | Gemini-Ultra Refiner     | 27.25 (+0.03)         | 22.05 (-1.91)         |
> | **GPT-4o Refiner**                  | **28.04 (+0.82)**     | **21.96 (-2.00)**     |
>
> Here, the baselines are:
> - SD v1.5 - The base Stable Diffusion model with the RealisticVision checkpoint.
> - No-Refiner: Use base author-provided descriptions from Civit.ai or Huggingface.
> - Gemini-Ultra Refiner: Use Gemini-Ultra as the Refiner’s VLM to generate better adapter descriptions. This is the version of Stylus presented throughout our paper.
> - GPT-4o Refiner: Use GPT-4o as the Refiner’s VLM.
>
> The quality of author-provided descriptions are poor, leading to worse performance than the typical SD checkpoint. Further improved refiner descriptions from GPT-4o can significantly boost Stylus’s performance, achieving the best textual alignment (CLIP) and image quality (FID) across all baselines, surpassing our original Stylus (with Gemini-Ultra).
>
> **Merging Adapters.** We clarify Eqn. 2 *averages* adapter weights per task and *sums* adapter weights across tasks. We take several measures below to ensure that the second term in Eqn. 2 does not grow too large:
> - Our masking scheme reduces the number of adapters in the final composition of LoRAs. (Sec 3.4)
> - Empirically, with the COCO dataset, we observed the composer identifies at most seven tasks with associated adapters. We also have the option in the composer’s prompt to limit the number of tasks (Tab. 2).
> - β scales down the rate at which the second term grows. We determined β=0.8 prevents highly-weighted adapters from overriding other concepts specified in the prompt, a challenge discussed in Fig. 13(b).
>
> We appreciate your feedback and believe these clarifications should address your concerns. We are open to further discussions to improve the paper. Thank you once again for your valuable insights!
>
> [1] Sheng, Ying, et al. S-LoRA: Serving Thousands of Concurrent LoRA Adapters. arXiv:2311.03285, arXiv, 5 June 2024. arXiv.org, https://doi.org/10.48550/arXiv.2311.03285.
>
> [2] Wu, Bingyang, et al. "dLoRA: Dynamically Orchestrating Requests and Adapters for LoRA LLM Serving." 18th USENIX Symposium on Operating Systems Design and Implementation (OSDI 24), USENIX Association, July 2024, pp. 911-927, Santa Clara, CA, www.usenix.org/conference/osdi24/presentation/wu-bingyang.

---

> > ### Comment · Reviewer_QQJ7 · 2024-08-11
> > **All of my concerns have been addressed**
> >
> > Thanks for the clarifications. All of my concerns have been addressed. I am raising the rating from weak accept to accept.

---

> > > ### Author Response · Authors · 2024-08-12
> > >
> > > We are glad that our rebuttal have addressed all your concerns and thank the reviewer for increasing the rating!

---

### Author Rebuttal · Authors · 2024-08-07

We are grateful to all the reviewers for their insightful feedback and enthusiastic reviews! To name just a few comments, reviewers acknowledged that Stylus is novel,

> This is a great paper. Original, high quality, clear, and significant. (Reviewer N5AD)

> This work presents a novel system, Stylus  (Reviewer wGTi)

> The idea is interesting … this paper investigates the adapters and to finalize a suitable model to generate good image given a fixed prompt. (Reviewer QQJ7)

timely and impactful,

> advances the field by providing a robust and automated solution (Reviewer wGTi)

> The use of adapters and PEFT will/should continue to increase in the future. (Reviewer N5AD)

especially for our open-source StylusDocs dataset,

> The creation of StylusDocs, a curated dataset with 75K adapters and pre-computed embeddings, adds substantial value to provide a rich resource for further experimentation and development.  (Reviewer wGTi)

and is comprehensively evaluated.

>Comprehensive Evaluation: The paper provides a thorough evaluation of Stylus across multiple metrics (CLIP, FID) and diverse datasets (Microsoft COCO, PartiPrompts). This robust evaluation framework enhances the credibility of the claimed performance improvements. (Reviewer wGTi)


___
___


We’d also like to highlight some of the shared feedback across reviewer rebuttals.

**Impact of Refiner (QQJ7, wGTi).** Our additional ablation experiments demonstrate that Stylus’s performance benefits significantly from the high quality of adapter descriptions provided by the VLM-based Refiner.

| Baselines (CFG=6)        | CLIP: ↑ is better     | FID: ↓ is better      |
|--------------------------|-----------------------|-----------------------|
| SD v1.5                  | 27.22                 | 23.96                 |
| No-Refiner               | 24.91 (-2.31)         | 24.26 (+0.3)          |
| Gemini-Ultra Refiner     | 27.25 (+0.03)         | 22.05 (-1.91)         |
| **GPT-4o Refiner**                  | **28.04 (+0.82)**     | **21.96 (-2.00)**     |

Here, the baselines are:
- SD v1.5: The base Stable Diffusion model with the RealisticVision checkpoint
- No-Refiner: Use base author-provided descriptions from Civit.ai or Huggingface.
- Gemini-Ultra Refiner: Use Gemini-Ultra as the Refiner’s VLM to generate better adapter descriptions. This is the version of Stylus presented throughout our paper.
- GPT-4o Refiner: Use GPT-4o as the Refiner’s VLM.

Without the refiner, the poor quality of author-provided descriptions results in Stylus performing worse than SDv1.5. However, the high quality of adapter descriptions from the GPT-4o Refiner results in the best performance, surpassing Gemini-Ultra Refiner, the original refiner VLM.

**Safety and Reliability of Adapter Descriptions (QQJ7, wGTi).** Stylus ensures adapter *safety* through a multi-stage filtering pipeline, initially excluding all explicitly tagged adapters by Civit.ai (Sec 3.2), followed by using Google's VertexAI API filters to reject unsafe adapters based on LLM-generated descriptions (Sec. A.7). For *reliability/accuracy,* Stylus's refiner uses example images from Civit.ai’s model card as a grounding mechanism to generate more accurate descriptions (Sec A.1), and we manually inspect and blacklist low-quality, explicit, or highly-inaccurate adapters (Sec A.4).

**Masking and Merging Clarification (QQJ7, rCTt, wGTi).** Reviewers asked for more clarity on Stylus’s masking and merging steps. Recall that the composer decomposes the prompt into tasks and maps highly relevant adapters to each task. For each task, a mask either selects A) just one of the task’s adapters, B) all of the task’s adapters, C) or none of the adapters. The selected adapters are then merged, with adapter weights *averaged* per task and then *summed* across tasks.

---

### Comment · Area_Chair_VXLT · 2024-08-12
**If you have any further questions for the authors, please ask them now**

Dear reviewers,

Thank you for your continued efforts in helping create the best possible NeurIPS 2024.

These reviews are very enthusiastic; I urge you still to check each others reviews and the rebuttals to check whether you have any further questions for the authors -- you have one more day to interact with them.

Your AC

---

### Decision · Program_Chairs · 2024-09-25

**Decision:**

Accept (oral)

**Comment:**

This paper explores systematically using adapters to improve the quality of diffusion model generated images, at reduced cost and increased efficiency. The paper studies how to automatically select adapters from a large library based on the given prompt.

The reviewers were all enthusiastic about the paper, and it would have a broad appeal to the NeurIPS audience. I therefore recommend an oral.